# OpenReview forum: "SELECTIVE FINE-TUNING FOR TARGETED AND ROBUST CONCEPT UNLEARNING"
_ICLR.cc/2026/Conference — Submitted to ICLR 2026_

### Official Review · Reviewer_LEwL · 2025-10-30

**Soundness:** 2
**Presentation:** 2
**Contribution:** 3
**Rating:** 6
**Confidence:** 3

**Summary:**

This paper proposes TRUST, a selective fine-tuning framework for concept unlearning in T2I tasks. The method has three parts: (1) discovering concept neurons by dynamically estimating masks in cross-attention via gradients of CLIP Score; (2) two unlearning objectives**—**CIP  and CSR ; and (3) an adaptive mask-guided fine-tuning process that repeatedly re-estimates masks during training. The method targets both single concepts and concept combinations. The paper claims better unlearning of individual and compositional concepts with smaller utility loss and improved efficiency.

**Strengths:**

- Handles concept combinations/conditional associations, which many unlearning baselines struggle with; TRUST reports effectiveness on CCE/CoCE while preserving individual benign concepts.
- Method-level clarity and modularity: dynamic, gradient-aligned discovery of concept neurons in CA (k/q/v) + two objectives (CIP & CSR ) that minimize, respectively, the number and the sensitivity of targeted parameters.
- Empirical utility preservation: on non-targeted concepts, small changes in ΔFID/CLIP/TIFA compared to the original model, while improving robustness vs. several adversarial prompt generators.

**Weaknesses:**

- CFG assumption & negative prompts. CFG is not universal (e.g., distilled models like SDXL-Turbo). Since UNet fine-tuning changes parameters shared by unconditional/negative branches, please qualify CFG-dependent claims and explain how TRUST behaves in non-CFG settings, including any mitigation for negative-prompt consistency (e.g., prompt dropout/rebalancing).
- Typo in Algorithm 2. The loss type is listed as {CR, CN}; main context indicates these should correspond to {CSR, CIP}. Please fix the notation and ensure consistency across text/algorithms.
- Sampling strategy & hyperparameters. State core choices in the main text: sampler (DDIM in the code),  whether gradients are accumulated across the full sampling trajectory, default value of the threshold $\xi$ (2.0 in the code).
- The presentation can be improved.

**Questions:**

See weakness

---

> ### Author Response · Authors · 2025-11-24
> **Response to Reviewer LEwL**
>
> We would like to thank the reviewer for their valuable and thoughtful feedback on our work and for their efforts to understand our work in depth. We are very happy to see that the reviewer recognises our effort towards introducing the problem of conditional unlearning. Thank you for identifying the important weakness in our work. We respond to the provided weaknesses as follows:
>
> ## Responses to the Weaknesses
> >**CFG assumption & negative prompts. CFG is not universal...**
> * We apologise for the misunderstanding with regards to CFG in L323-L325; here, the statement reflects the base model training behaviour and does not reflect TRUST's fine-tuning process. To clarify, TRUST fine-tuning uses conditional image generation via text-guided diffusion models, updating cross-attention projection weights directly. During inference, it's just a forward process.
> * In the case of unconditional generative models, TRUST's fine-tuning process relies on a subset of images with the concepts that are supposed to be unlearned, gradients from which will end up updating appropriate parameters according to the gradient values computed during the back propagation. Therefore, as TRUST does not have any dependence on CFG, we believe it should work effectively for non-CFG trained models as well.
>
> >**Typo in Algorithm 2...**
>
> * We thank the reviewer for identifying this typo. The final manuscript is updated accordingly.
>
> >**Sampling strategy & hyperparameters...**
> * We thank the reviewer for suggesting adding more implementation details to the main paper. Abiding by the suggestions, we have updated the main paper.
>
> >**The presentation can be improved...**
> * We request the reviewer to kindly share any issues with the presentation which we can improve upon. We are eager to improve the presentation to enable more clarity.
>
> Once again, we thank the reviewer for their insightful questions, and we hope that our responses will help clarify and address all the concerns raised and will make our work more acceptable to the research community at ICLR. Should there be any remaining concerns, questions or suggestions regarding our responses or the updated manuscript, we would be very grateful to the reviewers for sharing those with us.
>
> With Best Regards,
> The Authors

---

> > ### Comment · Reviewer_LEwL · 2025-11-27
> > **Reply to the rebuttal**
> >
> > Thanks to the authors for clarifying the questions I raised. Below are my follow-up comments:
> >
> > Non-CFG claim: The statement "should work effectively for non-CFG models" lacks empirical evidence. Please add a small non-CFG experiment.
> >
> > Figures (readability): Fig. 1–3 are dense and small. Fig. 3 (left, method overview) is tough to read. Please enlarge fonts, reduce panel density, and adjust spacing to protect readers' eyes:)

---

> > > ### Author Response · Authors · 2025-12-03
> > > **Response to the follow up comments**
> > >
> > > We thank the reviewers for their kind review of our responses. We answer to the follow-up comments as follows:
> > > >**Non-CFG claim: The statement "should work effectively for non-CFG models" lacks empirical evidence. Please add a small non-CFG experiment.**
> > > * To prove the validity of the TRUST framework across non-CFG models, we run TRUST on SDXL-Turbo (in the do_classifier_free_guidance = False setting). We observe similar results as SD 1.5, where the targeted concepts were unlearnt quickly (taking 50 steps with 21 seconds per step for CIP loss and 120 steps with 23 seconds per step for the CSR loss), while preserving the non-targeted concepts.
> > > * We have added the visual results for unlearning through both CIP and CSR losses under figure. Moreover, we also add supporting quantitative results for $\Delta FID$ scores. We get $\Delta FID_{CIP}=5.98$ and $\Delta FID_{CSR}=1.73$. From this, we can conclude that CIP loss performs more harsh unlearning. We believe that tuning the set of hyperparameters, especially $\beta_{CIP}$ could help resolve this issue.
> > >
> > > >**Figures (readability): Fig. 1–3 are dense and small.**
> > > * We will re-adjust the size of the figures to enable smooth reading in the camera-ready version.
> > >
> > > We hope that our newly added results for non-CFG models (SDXL Turbo) meet the requirements of the reviewer.

---

### Official Review · Reviewer_N5XW · 2025-10-30

**Soundness:** 2
**Presentation:** 2
**Contribution:** 3
**Rating:** 4
**Confidence:** 4

**Summary:**

Thiis paper proposes TRUST (Targeted Robust Selective fine-Tuning), a method for concept unlearning inT2I diffusion models. TRUST dynamically reestimates concept neurons inside cross-attention layers by measuring how gradients of a CLIP-based alignment objective respond to a target prompt. It then selectively fine-tunes only those neurons using one of two objectives: CIP (Concept Influence Penalty) which sparsifies high-impact neurons and CSR (Concept Sensitivity Reduction) which reduce gradient sensitivity (Hessian-aware soft suppression).  TRUST recomputes the mask at every step to track representation drift. Experiments report very low ASR against several adversarial prompt generators, small difference in the generation quality of retained concepts, support for concept combinations.

**Strengths:**

1- The contribution is clear. The paper identifies and acts on the fact that salient neurons shift during optimization, so a static mask is suboptimal.

2- The two finetuning objectives (CIP vs CSR) provide a useful trade-off between removal effectiveness and retention of other concepts.

3- Evaluation on multiple benchmarks and the consideration of multiple baselines.

4- Good results.

**Weaknesses:**

1- My main concern is that efficiency claims focus on steps, not compute. Dynamic mask recomputation requires repeated gradient passes. CLIP forward/backwards are required every step. The method relies on a generated image which means backpropagation through the VAE too. The paper compares steps but doesn’t report wall-clock, GPU hours, memory, or FLOPS vs. baselines tuned for speed. The 2.5 faster claim in abstract is misleading.

2- It's unclear whether the basline values are reproduced or reported based on the original papers. The paper does not mention the finetuning setting of the baselines. It's unclear whether the evaluations are fair.

3- The paper uses CLIP-based saliency and it also evaluates on CLIP. The mask definition is driven by CLIP Score gradients. Fidelity evaluation also leans on CLIP/TIFA. This can create objective-metric coupling: if the model learns to game CLIP features, both saliency and reported utility may look good while semantic unlearning in a human sense is weaker.

4- There is no study of the hyperparemters $\xi$ and $\beta$. How were the values in Appendix A.8 chosen?

5- CSR and CIP are not compared in wall-clock time and compute per step. There is an unclear "2-3 hours for CSR experiments and took 3-4 hours per epoch for CIP experiments." in line 1220 which in insufficient.

6- Do the considered baselines add a preservation loss to the finetuning objective? That is orthogonal to the unlearning objective.

**Questions:**

1- Why use CLIP for saliency? Have you tested saliency via alternative alignment signals (BLIP, QA loss, etc) and does performance hold? How correlated would masks be across aligners?

2-  Please report wall-clock time, Compute, peak memory, and the per-step overhead of mask recomputation (include CLIP backprop) vs. static-mask baselines based on noise prediction.

3- Line 409-410: "Notably, it achieves 18.52%, 12.1%, 10.57%, 14.33%, and 1.89% degradation in ASR over the UnlearnDiffAtk, MMA-Diffusion, Ring-a-Bell and P4D baselines." This doesn't make sense and should be revised.

4- Line 460-462 contradicts  Table 3. CSR users fewer steps. Is this a type?

---

> ### Author Response · Authors · 2025-11-24
> **Response to Reviewer N5XW (part 1)**
>
> We would like to thank the reviewer for their valuable and thoughtful feedback on our work and for their efforts to understand our work in depth. Thank you for identifying the important weakness in our work. We respond to the provided weaknesses as follows:
>
> ## Response to Weaknesses
> >**My main concern is that efficiency claims focus on steps, not compute...**
> * We completely agree with the reviewer and have updated Table 3 in the paper with the time taken(in hours) for all the discussed methods for unlearning a common concept. Moreover, we added Tables 5 and 8 in Appendix to present and compare the algorithm time and space complexity, and inference time and GPU storage, respectively, to add to the clarity of the method and actual time taken for editing the model.
>
> >**It's unclear whether the basline values are reproduced or reported based on the original papers...**
> * The baseline results reported in Table 2 are borrowed from the original papers. However, metrics such as \delta FID, CLIP, and TIFA are recomputed over a consistent set of safe prompts from the COCO dataset.
>
> >**The paper uses CLIP-based saliency and it also evaluates on CLIP...**
> * The TIFA metric accesses the fidelity of the T2I generative models leveraging a VQA(mPLUG-owl 3) model and an LLM(LLaMa), which are therefore not biased by the saliency map computation on the CLIP alignment. Moreover, we also compute and report the Unlearning Accuracy(UA) and Retaining Accuracy(RA), which shows the superior ability of our technique over existing unlearning techniques.
>
> >**There is no study of the hyperparemters...**
> * We have updated the manuscript section A.11.3 with the ablations on the hyperparameter $\eta$. Furthermore, we have added more supporting arguments for the choice of hyperparameters $\beta$ under A.11.2. Overall, from the ablation, we conclude that the value of the hyperparameter $\beta$ was chosen to keep the loss computed using CIP and CSR comparable to the preservation loss and to also keep the overall loss value in the range of 0.1-0.01 for effective finetuning. The ablation study on the hyperparameter $\xi$ clearly indicates that $\xi = 2$ maintains a sweet spot between the total number of fine tuning steps and overall accuracy.
>
> >**CSR and CIP are not compared in wall-clock time and compute per step...**
> * We have updated section A.8 with time (in seconds)  per step required for the CSR and CIP loss functions for clarity. Each finetuning step for CIP and CSR took 23 and 16 seconds, respectively, on average.
>
> >**Do the considered baselines add a preservation loss to the finetuning objective?**
> * We have updated Table 2 with a column indicating the techniques which use a preservation component.

---

> ### Author Response · Authors · 2025-11-24
> **Response to Reviewer N5XW (part 2)**
>
> ## Responses to Questions
>
> We thank the reviewer for their important and interesting questions and respond to them as follows:
> >**Why use CLIP for saliency?**
> * We use the CLIP score for computing the saliency map because of its well-known ability to effectively capture the relationship between the images and corresponding text by computing the embeddings for both of them in the same latent space. Furthermore, since the text encoder in Stable Diffusion 1.5 is itself a CLIP-based model, employing a CLIP-derived alignment loss is both natural and optimally aligned with the underlying architecture, and is therefore used in multiple prior works [1,2,3]. The noise-based saliency map computation is discussed by SalUn, whose results are extensively compared against in the paper. We have added the motivation regarding choosing CLIP score for computing saliency maps under section A.11.1 in the Appendix.
> * However, we acknowledge that other alignment scores, such as the BLIP score or the VQA-based score, could be leveraged for identifying the saliency maps. We limit the focus of this study to using the CLIP alignment score for this purpose, considering the above-stated reasons.
>
> >**Please report wall-clock time, Compute, peak memory...**
>
> * As suggested by the reviewer, we have updated Table 3 in the paper with the time taken(in hours) for all the methods for unlearning a common concept.
> * Moreover, we added Tables 5 and 8 in the Appendix to present and compare the algorithm time and space complexity, and inference time and GPU storage, respectively, to add to the clarity of the method and actual time taken for editing the model.
>
> >**Line 409-410: "Notably, it achieves...**
> * The percentage improvement discussed in lines 409-410 is computed by subtracting the best ASR performance of the existing methods from our method to highlight the improvement over the set of adversarial prompting techniques.
>
> >**Line 460-462 contradicts Table 3. CSR users fewer steps. Is this a type?**
> * We thank the reviewer for identifying this typo. The final manuscript is updated accordingly.
>
>
> Once again, we thank the reviewer for their insightful questions and we hope that our responses would help in clarifying and addressing all the concerns raised and would make our work more acceptable by the research community at ICLR. Should there be any remaining concerns, questions or suggestions regarding our responses or the updated manuscript, we would be very grateful to the reviewers for sharing those with us.
>
> With Best Regards
> The Authors
>
> ---
>
> [1] Dongzhi Jiang, Guanglu Song, Xiaoshi Wu, Renrui Zhang, Dazhong Shen, Zhuofan Zong, Yu Liu, and Hongsheng Li. Comat: Aligning text-to-image diffusion model with image-to-text concept matching. In The Thirty-eighth Annual Conference on Neural Information Processing Systems, 2024. URL https://openreview.net/forum?id=OW1ldvMNJ6.
> [2] Zikui Cai, Yaoteng Tan, and M. Salman Asif. Targeted unlearning with single layer unlearning
> gradient. In Neurips Safe Generative AI Workshop 2024, 2024. URL https://openreview.
> net/forum?id=ePKuQQwCGm.
> [3] Gwanghyun Kim, Taesung Kwon, and Jong Chul Ye. Diffusionclip: Text-guided diffusion models for robust image manipulation. In 2022 IEEE/CVF Conference on Computer Vision and Pattern Recognition (CVPR), pp. 2416–2425, 2022. doi: 10.1109/CVPR52688.2022.00246.

---

> > ### Comment · Reviewer_N5XW · 2025-11-27
> >
> > I thank the authors for their response. They have addressed some of my concerns.
> >
> > ### Efficiency Claims
> > The experimental setup used to compute the latency metrics in table 3 (for both TRUST and the baselines) is currently missing. Is the TRUST setting identical to A.8? Are baselines also using batch size > 1? It is unclear how these values were obtained. In addition, the discussion of the method’s speed (e.g., lines 466–475 or lines 67-68) requires revision to accurately reflect the wall clock time needed for experiments rather than number of steps.
> > The efficiency metrics in Table 8 offer limited insight because the baseline models and TRUST were evaluated on different hardware. In addition, only inference metrics are reported. There is no comparison of training memory usage. Since each TRUST step involves backpropagation through multiple models, its training memory footprint should be included. This represents a genuine limitation of the approach, and it should be stated clearly that TRUST may require GPUs with substantially larger memory, rather than omitting relevant information.

---

> > > ### Author Response · Authors · 2025-12-03
> > > **Response to the reviewer on efficiency claims**
> > >
> > > We would like to thank the reviewer for their extended questions. We address them as follows:
> > >
> > > >**Is the TRUST setting identical to A.8?**
> > > * As correctly pointed out by the reviewer, the setting of TRUST is identical to A.8. For the other baselines, we use the same settings as recommended in their published papers.
> > >
> > > >**Are baselines also using batch size > 1?**
> > > * Yes, the baselines have batch sizes >1, except for ESD, whose batch size is set to 1 in their concerned published papers. For reference, the batch sizes of the used models are as follows:
> > > * Retrain -> 128
> > > * SalUn -> 128
> > > * CA -> 8
> > > * CoGFD -> 20
> > > * Ours (both CIP and CSR) -> 5
> > >
> > > >**In addition, the discussion of the method’s speed (e.g., lines 466–475 or lines 67-68) requires revision**
> > > * We have updated the discussion in lines 466-475 and lines 67-68 to add the wall clock time as well.
> > >
> > > >**The efficiency metrics in Table 8 offer limited insight because..**
> > > * We agree that it might not be wise to compare the inference time and memory usage as they directly depend on the underlying hardware; however, the overall accuracy of unlearning should suggest meaningful insights.
> > >
> > > >**There is no comparison of training memory usage**
> > > * We have updated section A.8 with the peak memory usage for both CIP and CSR training. The peak memory usage of CIP is 13.6 GB, and that of CSR is 50 GB for batch size = 5 for SD 1.5. The peak memory usage of CIP is 39 GB (approximately), and that of CSR is  75.5 GB for batch size = 2, while training SDXL Turbo. All the training is done on one A100 GPU with 80 GB of memory.
> > > * We believe that this is quite decent memory usage for both the loss functions, and hope would further clarify the exact requirements for fine-tuning using TRUST.
> > >
> > > We hope that our responses clarify the memory and runtime requirements of TRUST more clearly and address all the concerns raised by the reviewer.

---

### Official Review · Reviewer_re6D · 2025-10-31

**Soundness:** 3
**Presentation:** 3
**Contribution:** 3
**Rating:** 6
**Confidence:** 3

**Summary:**

The paper proposes TRUST, which achieves selective unlearning of specific concepts and concept combinations, while preserving the quality and efficiency for benign concepts.  Specifically, TRUST first dynamically estimates “concept neurons” within cross-attention layers and repeatedly updates the mask during fine-tuning so that only those parameters are selectively adapted. In addition, two complementary regularizers are introduced: CIP (Concept Influence Penalty), which performs “hard” unlearning by penalizing the cardinality of high-influence parameters; and CSR (Concept Sensitivity Reduction), which performs “soft” unlearning by minimizing the gradient sensitivity of the noise-prediction to the target parameters. Extensive experiments demonstrate the effectiveness of the proposed method.

**Strengths:**

* The writing is fluent and logically coherent, exhibiting strong readability.
* Dynamic localization of concept neurons mitigates drift. TRUST re-estimates the mask each step, avoiding outdated static selections and directly addressing observed “saliency drift” during training.
* Complementary regularizers for hard/soft unlearning. CIP and CSR cover different deployment needs (compliance-oriented vs. fidelity-oriented) and are accompanied by clear mechanistic contrasts and visual analyses.
* Strong adversarial robustness with minimal collateral damage. ASR drops significantly under diverse attacks while benign-prompt quality changes (e.g., $\Delta$FID/CLIP/TIFA) remain small, indicating a well-balanced forget/retain trade-off.

**Weaknesses:**

* Some ablation studies are needed to demonstrate the effectiveness of the method. For example, in the case of the CIP regularization, how does it compare to directly deactivating all concept neurons?
* It is necessary to show more diverse visual examples of concept unlearning, for example, removing specific stylistic concepts.
* For the conditional concept unlearning problem, is there any relationship between the distribution of activated neurons and that of single-concept unlearning? For example, do the neurons corresponding to “a cat on the table” represent the intersection of those associated with “cat” and “table”?

**Questions:**

See 'Weaknesses'.

---

> ### Author Response · Authors · 2025-11-24
> **Response to Reviewer re6D**
>
> We would like to thank the reviewer for their valuable and thoughtful feedback on our work and for their efforts to understand our work in depth. We are really happy to see that the reviewer appreciates the adversarial robustness of TRUST and the in-depth exploration of the CIP and CSR regularizers. Thank you for identifying the important weakness in our work. We respond to the provided weaknesses as follows:
>
> ## Responses to Weaknesses
> >**Some ablation studies are needed to demonstrate the effectiveness of the method...**
> * We found this idea by the reviewer intriguing and therefore, conducted the requested experiment. We have updated the appendix (section A.12) with the experiments on deactivating the identified concept neurons. We deactivate (zero the activation) for the identified concept neurons to check if they were exclusively responsible for expressing the concept. We do this for both artistic styles (eg. Van Gogh) and for concrete concepts (eg. Dog).
> * Through our experiments, we can roughly conclude that simple deactivation of the identified set of concept neurons only helps with preventing the model from generating subtle artistic styles like Van Gogh, but is quite ineffective for other concrete concepts like “Dog”. Moreover, this deactivation leads to impacting the generation quality of the non-targeted concepts.
>
> >**It is necessary to show more diverse visual examples of concept unlearning...**
> * As TRUST targets on removing any abstract concept, we did not experiment with removing stylistic patterns or artistic styles. However, to address the issues raised by the reviewer, we present the results on removing some famous artistic styles by leveraging the TRUST framework under section A.10. We also present the results of unlearning artistic styles on the UnlearnCanvas dataset [1] under Table 8 in the updated manuscript.
> * From our results, we conclude that TRUST could indeed be used for unlearning artistic styles as well. Moreover, we observe that unlearning artistic styles is rather simpler as it requires comparatively fewer finetuning steps for unlearning the target artistic style.
>
> >**For the conditional concept unlearning problem, is there any relationship between the distribution..**
> * As pointed out by the reviewer, we do observe an intersection in the concept neurons of individual concepts and that of the combination of concepts. Figure 19 (in the updated paper) highlights this overlap by the edges between the concepts, where the weight of the edges is the sum of the gradients computed corresponding to the number of common concept neurons between the two related concepts.
> * From the figure, we can conclude that the combined concept: “Cat on a Table” shares 7 and 5 concept neurons with the constituent concepts “Cat” and “Table”, respectively. We are currently studying this new concept knowledge graph, to develop more fine-grained unlearning techniques.
>
> Once again, we thank the reviewer for their insightful questions, and we hope that our responses will help clarify and address all the concerns raised and make our work more acceptable to the research community at ICLR. Should there be any remaining concerns, questions or suggestions regarding our responses or the updated manuscript, we would be very grateful to the reviewers for sharing those with us.
>
> With Best Regards, The Authors
>
> ---
> [1] Yihua Zhang, Chongyu Fan, Yimeng Zhang, Yuguang Yao, Jinghan Jia, Jiancheng Liu, Gaoyuan Zhang, Gaowen Liu, Ramana Kompella, Xiaoming Liu, and Sijia Liu. Unlearncanvas: a stylized image dataset for enhanced machine unlearning evaluation in diffusion models. In Proceedings of the 38th International Conference on Neural Information Processing Systems, NIPS ’24, Red Hook, NY, USA, 2024a. Curran Associates Inc. ISBN 9798331314385.

---

> > ### Comment · Reviewer_re6D · 2025-11-27
> >
> > Thank you. The authors’ response has addressed the concerns I raised, and I will maintain my positive score.

---

### Official Review · Reviewer_SzsB · 2025-10-31

**Soundness:** 3
**Presentation:** 3
**Contribution:** 3
**Rating:** 6
**Confidence:** 1

**Summary:**

This work proposes TRUST, a new framework for concept unlearning in text-to-image models by dynamically identifying concept neurons using cross-attention saliency.  The authors also introduce two new unlearning objective functions that dynamically update the identified neurons. Experimental results show that TRUST is robust against adversarial prompts while preserving generation quality.

**Strengths:**

I am not quite familiar with unlearning for diffusion models, and therefore cannot confidently assess the quality of this paper. I would recommend that the AC seek input from reviewers who are more familiar with this topic.

**Weaknesses:**

Line 53: there should be a comma before "leading to".

Line 107: missing space in TRUSTis.

Line 194: suppress $\to$ suppresses

**Questions:**

N/A

---

> ### Author Response · Authors · 2025-11-24
> **Response to Reviewer SzsB**
>
> We would like to thank the reviewer for their valuable time and on their efforts to understand our work. Thank you for identifying the editing issues in our work. We have updated the current manuscript with the recommended formatting changes.
>
> With Best Regards,
> The Authors

---

### Official Review · Reviewer_VqNF · 2025-11-03

**Soundness:** 3
**Presentation:** 3
**Contribution:** 3
**Rating:** 4
**Confidence:** 4

**Summary:**

This work introduces a saliency-based unlearning algorithm for text-to-image diffusion models, which:
1) considers the dynamic nature of salient neuron localization during the unlearning process;
2) achieves promising empirical performance in unlearning compared to state-of-the-art baselines, while demonstrating strong preservation of non-target concepts and computational efficiency;
3) introduces challenging unlearning tasks (concept combination and conditional concept unlearning) to demonstrate the precision of the proposed method.

The claimed benefits are supported primarily by empirical experiments and demonstrations, with no theoretical justification provided.

**Strengths:**

- Strong motivation to build upon the salient parameter shifts observed during the fine-tuning process.

- Strong demonstration of empirical improvement.

- Introduces conditional concept unlearning, which serves as a strong test of unlearning effectiveness at the sentence semantic level.

**Weaknesses:**

- **Missing relevant work in discussions.** This work proposes a saliency-based method. While SalUn [1] is thoroughly discussed, other relevant saliency-based methods [2][3][4] are neither discussed nor compared. In particular, [4] utilizes a loss design on CLIP alignment for saliency parameters that is similar to TRUST (the proposed method).

[1] Fan et al., Salun: Empowering machine unlearning via gradientbased weight saliency in both image classification and generation. ICLR, 2024

[2] Foster et al., Fast machine unlearning without retraining through selective synaptic dampening. AAAI, 2024

[3] Dong et al.,Towards safe concept transfer of multimodal diffusion via causal representation editing. NeurIPS, 2024

[4] Cai et al., Targeted Unlearning with Single Layer Unlearning Gradient. ICML 2025.


- **Erasure efficiency claim.** The authors try to justify runtime efficiency mainly through number of fine-tuning steps (Line 460-467). However, this is not a straight forward metric and potentially misleading, as fine-tuning iteration can be designed in various ways. For example, TRUST (the proposed method) involves iterative salient mask, loss gradient computations and model update in the loop, vs SalUn does one-time salient mask computation outside the loop, and compute gradient and update model iteratively. I suggest author to measure computational efficiency in more straight forward units (e.g., seconds), or provide a summary table on  1-iter runtime for each method in the Table 3.


- **Lacks ablations.** This work empirically demonstrated the improved performance with key designs lie in the following aspects:
(1) utilizing CLIP alignment of target data as forget loss for iterative salient-mask computation;
(2) Loss terms $\mathcal{L}\_\mathrm{CIP,CSR}$ that has loss
$\mathcal{L}\_\mathrm{prev}$ from SalUn (Fan et al., 2024)  .
It would be beneficial if author can included ablation experiments on these key designs to reveal which design played a crucial role.
Besides, like SalUn, the proposed method also relies on a introduced hyperparameter for thresholding saliency parameter ($\gamma$ in Eqn. 5), the effect of this hyperparameter is also not discussed or studied.


- **Writing clarity.** Line 328-331 relation of $\mathcal{L}_\mathrm{CSR}$ to "second-order information", "underlying Hessian" is hand wavy, need further explaination/derivation. Also the "Appendix-F" appeared in Line-331 is not properly cross-referenced.


- **Experimental models are outdated and scalability of proposed method is questionable.** The main experiments employ Stable Diffusion v1.5 (released in Q4 2022, params < 1B) out of comparison consistency with state-of-the-art baselines. This model is built on a UNet-based denoiser architecture, which is now outdated, as the latest text-to-image models have adopted DiT or MMDiT denoiser architectures. These transformer-based designs scale more effectively and deliver higher image generation quality. As the main emphasis of this paper is on UNet-based text-to-image diffusion models, it is questionable whether the proposed method can be extended to more recent and advanced models that are more widely used in practice (e.g., Stable Diffusion 3, FLUX, SANA, Qwen-Image).

**Questions:**

- **Clarification on the term "neurons".** Does the author refer to the number of model parameters (i.e., the entries of a weight matrix)?




- **Saliency shift effect characterization.** This work measures the "saliency shift" effect primarily by counting the number of identified "concept neurons," which provides a collapsed view of high-dimensional data. Do the locations of these "concept neurons" also shift across the UNet?




- **Clarification on $\mathcal{L}_\mathrm{CSR}$.** What is the relation of $\mathcal{L}_\mathrm{CSR}$ to "second-order information", "underlying Hessian"? How does it provide more effective optimization for unlearning objectives?



- Refer to weaknesses for other questions/suggestions.

---

> ### Author Response · Authors · 2025-11-24
> **Response to Reviewer VqNF (part 1)**
>
> We would like to thank the reviewer for their valuable and thoughtful feedback on our work and for their efforts to understand our work in depth. We are very happy to see that the reviewer recognises our effort towards introducing the problem of conditional unlearning. Thank you for identifying the important weakness in our work. We respond to the provided weaknesses as follows:
> ## Responses to weaknesses
> > **Missing relevant work in discussions. This work proposes a saliency-based method...**
>
> * Thank you for pointing out other saliency-based unlearning approaches, which we inadvertently missed during our literature review phase. We are updating the paper with the discussion and comparison against these works, and would request the reviewer to kindly go through the updated manuscript.
> * However, we noticed that [2] was not developed for unlearning in the domain of T2I diffusion model, but the technique is interesting and quite similar to ours for the identification of saliency, except the fact that they use the Fisher Information Matrix to gauge the sensitivity of the model’s parameters against the computed loss. Instead, we leverage the gradients to measure the sensitivity (i.e. the first order derivative instead of the second order). Nonetheless, we have added the discussion to this interesting technique in our Related Works section and have compared it against other methods in Table 1.
>
> > **Erasure efficiency claim. The authors try to justify runtime efficiency mainly through number of fine-tuning steps...**
>
> * We completely agree with the reviewer and have updated Table 3 in the paper with the time taken(in hours) for all the methods for unlearning a common concept.
> * Moreover, we added Tables 5 and 8 in the Appendix to present and compare the algorithm time and space complexity, and inference time and GPU storage, respectively, to add to the clarity of the method and actual time taken for editing the model.
> * Also, we report the time taken per iteration for CIP and CSR losses. We observe that one iteration of fine-tuning for CIP takes 23 seconds and CSR takes 16 seconds, on average.
>
> > **Lacks ablations. This work empirically demonstrated the improved...**
> * We use the CLIP score for computing the saliency map because of its well-known ability to effectively capture the relationship between the images and corresponding text by computing the embeddings for both of them in the same latent space. Furthermore, since the text encoder in Stable Diffusion 1.5 is itself a CLIP-based model, employing a CLIP-derived alignment loss is both natural and optimally aligned with the underlying architecture, and is therefore used in multiple prior works [3,4,5]. The noise-based saliency map computation is discussed by SalUn, whose results are extensively compared against in the paper. We have added the motivation regarding choosing CLIP score for computing saliency maps under section A.11.1 in the Appendix.
> * However, we acknowledge that other alignment scores, such as the BLIP score or the VQA-based score, could be leveraged for identifying the saliency maps. We limit the focus of this study to using the CLIP alignment score for this purpose, considering the above-stated reasons.
> * Also, we have updated the manuscript with the ablations on the hyperparameter $\xi$, and have stated the grounding and reasoning for the selection of the hyperparameter $\beta$ under sections A.11.3 and A.11.2, respectively in the Appendix.
> * Through the ablations conducted for the hyperparameter $\xi$, we find the most optimal value of $\xi=2$ to achieve the best balance between the number of finetuning steps and the overall accuracy.
> * We choose $\beta=1e-3$ to keep the final loss in the optimal range for effective finetuning i.e. (0.1-0.01), and also comparable to the preservation loss.
>
> > **Writing clarity. Line 328-331 relation of...**
> * We have updated the manuscript regarding all the concerns regarding the writing clarity.
>
> > **Experimental models are outdated and scalability of proposed method is questionable...**
> * Past approaches[1] have shown that major concept-related information lies in the cross-attention layers of the Unet-based denoiser for T2I diffusion models. Therefore, TRUST exclusively works with the layers possessing maximum concept-related information(cross-attention in case of Unet-based denoiser systems). As correctly pointed out by the reviewer that the very recent T2I diffusion models have shifted to DiT and MMDiT denoisers, which usually don’t have the cross-attention layers. However, recent works[2] have similarly proved the existence of concept-related information in the self-attention layers of these transformers. Therefore, we expect TRUST to similarly work for the recent denoisers by isolating concept neurons within the self-attention layers. We chose SD 1.5 for our evaluation to show the effectiveness of unlearning towards potentially harmful concepts, as the model was trained on harmful data.

---

> ### Author Response · Authors · 2025-11-24
> **Response to Reviewer VqNF (part 2)**
>
> ## Responses to Questions
> We thank the reviewer for their important and interesting questions and respond to them as follows:
> >**Clarification on the term "neurons"...**
> * By “neurons” we mean the key, value and query projection weight matrices in the cross attention layers of the denoiser.
>
> >**Saliency shift effect characterization...***
> * The saliency shift is observed in both the location as well as in the number of concept neurons identified. We have updated the paper with this to make it clearer.
>
> >**Clarification on $\mathcal{L}_{CSR}$...**
> * $\mathcal{L}_{CSR}$ aims towards flattening the gradient at the current weight of the tunable parameters. For this, the aim is to minimise the computed gradient. We use Hessian to approximate the computation of the double derivative using the first order logic twice. We have updated the section Appendix A.7 with this to aid in clarity.
>
> Once again, we thank the reviewer for their insightful questions, and we hope that our responses will help clarify and address all the concerns raised and make our work more acceptable to the research community at ICLR. Should there be any remaining concerns, questions or suggestions regarding our responses or the updated manuscript, we would be very grateful to the reviewers for sharing those with us.
>
> With Best Regards,
> The Authors
>
> ---
>
> [1] Nupur Kumari, Bingliang Zhang, Sheng-Yu Wang, Eli Shechtman, Richard Zhang, and Jun-Yan Zhu. Ablating concepts in text-to-image diffusion models. In 2023 IEEE/CVF International Conference on Computer Vision (ICCV), pp. 22634–22645, 2023a. doi: 10.1109/ICCV51070.2023.02074.
>
> [2] Helbling, Alec and Meral, Tuna Han Salih and Hoover, Benjamin and Yanardag, Pinar and Chau, Duen Horng. Concept Attention: Diffusion Transformers Learn Highly Interpretable Features. Forty-second International Conference on Machine Learning (2025) (oral).
>
> [3] Dongzhi Jiang, Guanglu Song, Xiaoshi Wu, Renrui Zhang, Dazhong Shen, Zhuofan Zong, Yu Liu, and Hongsheng Li. Comat: Aligning text-to-image diffusion model with image-to-text concept matching. In The Thirty-eighth Annual Conference on Neural Information Processing Systems, 2024. URL https://openreview.net/forum?id=OW1ldvMNJ6.
>
> [4] Zikui Cai, Yaoteng Tan, and M. Salman Asif. Targeted unlearning with single layer unlearning
> gradient. In Neurips Safe Generative AI Workshop 2024, 2024. URL https://openreview.
> net/forum?id=ePKuQQwCGm.
>
> [5] Gwanghyun Kim, Taesung Kwon, and Jong Chul Ye. Diffusionclip: Text-guided diffusion models for robust image manipulation. In 2022 IEEE/CVF Conference on Computer Vision and Pattern Recognition (CVPR), pp. 2416–2425, 2022. doi: 10.1109/CVPR52688.2022.00246.

---

### Author Response · Authors · 2025-11-24
**Common response and edits made in the updated manuscript**

## Edits made in the updated manuscript
We would like to thank the reviewer for their valuable and thoughtful feedback on our work and for their efforts to understand our work in depth. Thank you for identifying the important weakness in our work. We respond to each of the reviewers individually, addressing their questions and concerns. As a result of the rebuttal, we have added the following new set of experiments to the updated manuscript of the paper to further add clarity, in accordance with the suggestions received by the reviewers:
* Reporting of the Training time, Train-time storage, Inference time, Inference time storage, Train and inference-time overall time and storage complexity of our method (using both CIP and CSR) against other prominent works. Kindly refer to section A.8, A.9, A.2, Tables 3, 5, and 8 in the updated manuscript.
* Evaluation on unlearning artistic styles leveraging the Unlearn Canvas dataset. We report the Overall accuracy of unlearning (i.e. the average of Unlearning Accuracy and Retaining Accuracy) on unlearning artistic styles and also present visual results. Kindly refer to section A.10, Table 8, and Figure 14.
* Ablation studies to establish the need for the Preservation Loss, and for the optimal value of the hyperparameter $\xi$. Kindly refer to sections A.11.3, A.11.4, Table 9, Figure 15.
* Grounding and reasoning for choosing CLIP Score for concept neuron identification, and for the value chosen for the hyperparameter $\beta$. Kindly refer to sections A.11.1 and A.11.2 in the Appendix.
* Ablation study on simply zeroing out(shutting off) the activations for the identified set of concept neurons. Kindly refer to section A.12, and Figures 16, 17, and 18 for some visual results.
* Demonstration of overlap in the concept neurons shared by related concepts, highlighting neuron sharing, and validating superposition. Kindly refer to section A.13 and Figure 19.
* Results from performing unlearning on SDXL Turbo using the TRUST framework using both CIP and CSR loss functions. Kindly refer to section A.14 and Figure 20.

All the updates in the new manuscript are highlighted in Blue colour. We would be updating this common response throughout the rebuttal period, should we add any new experiments or ablations, to make it easy for the reviewers to track the modifications made in the paper.

---

### Author Response · Authors · 2025-12-03
**Summary of the Discussion Period**

Dear Area Chair,

It was really unfortunate to see the reviewer-author anonymity being compromised. We are grateful for your and the reviewers' service to the scientific community. We have addressed all the concerns raised by the reviewers and summarise the overall discussion period as follows:

Reviewers consistently highlight that our paper makes a clear, well-motivated, and practically impactful contribution to concept unlearning in text-to-image diffusion models. The core strengths identified across reviews were:

* **Dynamic concept-neuron localization:** TRUST addresses a key limitation of prior work by showing that salient neurons drift during optimization and by re-estimating concept neurons at every step. Reviewers praised this as a principled and empirically validated improvement over static masking.

* **Complementary unlearning objectives (CIP & CSR):** The introduction of two targeted objectives: CIP for hard, and CSR for soft, unlearning was noted as a valuable, modular design that enables different trade-offs between forgetting and retention.

* **Strong empirical performance and robustness:** Multiple reviewers emphasized that TRUST achieves state-of-the-art unlearning effectiveness, particularly under adversarial prompts, while maintaining high generation quality and minimal collateral damage on benign or unrelated concepts.

* **Handling concept combinations and conditional associations:** Reviewers agreed that TRUST is among the first methods to effectively unlearn compositional or conditional concepts (CCE/CoCE tasks), a setting where prior methods struggle. This was viewed as a strong demonstration of practical relevance.

During rebuttal, we have carefully addressed all the concerns raised by the reviewers, including core concerns: (1) Clarifications on the wall time, memory usage of TRUST, and (2) Ablations on some hyperparameters.
Apart from these, the discussion with the reviewers during the rebuttal period led to the following contributions to the work:
* **Reviewer VqNF** recognized some missing prior works, discussions on which were afterwards effectively integrated into the paper. The major concerns raised were for reporting the efficiency in wall clock time and performing ablations for the hyperparameters, which we fully addressed both in the comments and the updated paper. Unfortunately, the reviewer was not able to respond to our clarifications in time.
* **Reviewer re6D** raised an interesting question of deactivating the identified concept neurons. To address the concern, we thoroughly conducted experiments by deactivating the concept neurons and added qualitative results and discussion in the updated manuscript.
They also asked for a more diverse set of unlearning over artistic styles, which we addressed through extended experiments and discussions in the paper.
Finally, they also asked regarding the overlap of concept neurons across concepts, for which we constructed the overlap graph for better visualization of the overlap.
In response, the reviewer was satisfied with our experiments with no further questions.
* **Reviewer N5XW** raised major concerns regarding the runtime of the finetuning and memory usage. We thoroughly addressed all the concerns both in the first and the follow-up responses, by updating the paper with clear runtime and memory statistics, along with the algorithmic complexity, and per-step time taken for both the loss functions. We also clearly mentioned the setting for our fine-tuning as well as for the baselines in our responses and the updated manuscript. We are confident that we have addressed all the concerns of the reviewer.
* **Reviewer LEwL** questioned if TRUST could be applied to non-CFG models, and requested experiments to establish the same. Addressing their concerns, we conducted experiments on SDXL-Turbo(as requested) to establish that TRUST performs effective unlearning irrespective of CFG. We are confident that our experiments address the last concern raised by the reviewer.

All the edits made in the final manuscript as a part of the rebuttal phase discussions are highlighted in blue font.

Thank you so much for your time, efforts and service to the community.

With Best Regards

The Authors

---

### Meta-Review · Area_Chair_me8y · 2026-01-05

**Summary:**

This paper proposes TRUST, a selective fine-tuning framework for concept unlearning in text-to-image diffusion models that dynamically re-identifies concept-relevant neurons and applies targeted unlearning objectives. Reviewers broadly agreed that the core idea –– addressing saliency drift via dynamic neuron localisation –– is well motivated and technically sound, and that the method demonstrates strong empirical performance on standard unlearning benchmarks, including adversarial prompting and compositional concepts.

However, the reviews also raised substantial concerns about the empirical evaluation's practical efficiency, scalability, and clarity, particularly regarding compute and memory costs, the fairness of baseline comparisons, and the reliance on CLIP-based objectives and metrics. While the rebuttal added significant additional experiments and clarifications, the remaining concerns leave uncertainty about the method’s real-world practicality and the strength of the claimed efficiency advantages. These considerations informed the final recommendation.

**Reviewer Concerns:**

**Concerns largely addressed by the rebuttal:**

* Missing ablations (hyperparameters, preservation loss, neuron deactivation baselines).
* Lack of wall-clock runtime reporting and per-step timing.
* Applicability to non-CFG models (addressed via SDXL-Turbo experiments).
* Requests for additional qualitative results (artistic styles, concept overlap).
* Several clarity and presentation issues.

**Concerns partially or not fully resolved:**

* **Efficiency and scalability**: Although wall-clock time and peak memory were reported, TRUST still incurs substantially higher training-time memory and computational overhead than static-mask baselines, weakening the original efficiency claims.
* **Fairness and transparency of baseline comparisons**, including differences in batch size, hardware, and training setups.
* **Metric coupling risk**, as CLIP is used both for saliency estimation and for several evaluation metrics.
* Limited evidence that the approach will scale cleanly to newer large-scale transformer-based diffusion models beyond the tested settings.

**Reviewer Scores:**

Reviewer VqNF:
Initially 4 (weak reject). After the rebuttal, no explicit score update was provided.

Reviewer N5XW:
Initially 4 (weak reject). Acknowledged improvements after rebuttal, but continued to flag efficiency, memory, and comparison issues. Score likely remained 4.

Reviewer re6D:
Initially 6 (weak accept). Explicitly stated that the rebuttal addressed concerns and maintained the score.

Reviewer LEwL:
Initially 6 (weak accept). After adding non-CFG experiments and clarifications, maintained the score.

Reviewer SzsB:
6, but with very low confidence and limited domain expertise. Score unchanged.

---

### Decision · Program_Chairs · 2026-01-26

Reject